# Knowledge Distillation for Large Language Models through Residual Learning

**Thinh On**[1,2,*,†]**, Hengzhi Pei**[1]**, Leonard Lausen**[1,*]**, George Karypis**[1]
[1]Amazon Web Services, [2]New Jersey Institute of Technology

## Abstract

Knowledge distillation has become a crucial technique to transfer the capacities of large language models (LLMs) to smaller, more efficient models for practical deployment. While recent work exploits rich information from intermediate states of the teacher model for more effective knowledge transfer, imperfect knowledge from the teacher can also mislead student learning, restricting the student's generalization capacity. In this work, we propose a two-stage distillation framework that is effective for diverse knowledge distillation scenarios. In the first stage, we pretrain projectors to extract and compress teacher knowledge into a low-dimensional vector space via self-reconstruction. In the second stage, we perform distillation with a hybrid objective that combines learning from the compressed teacher representations with standard supervised fine-tuning on ground-truth data. Our key innovation is *residual learning* for LLM distillation, where the student learns to make predictions based on the differential between its representations and projected states from the teacher. This approach encourages the student to further improve its representations beyond potentially erroneous teacher knowledge. For Mixture-of-Experts (MoE) teacher models, we further fuse the experts' outputs using a self-attention mechanism for better utilizing the teacher knowledge. Moreover, to support the cross-tokenizer distillation setting, where the teacher and student models have different vocabularies, we adopt a cross-model attention mechanism that eliminates the need for explicit token alignment rules. Experimental results show the superior performance of our proposed framework under both same- and cross-tokenizer settings, demonstrating the effectiveness in preserving teacher knowledge and improving student generalization capability.

## 1 Introduction

Large language models (LLMs) have demonstrated remarkable capabilities across diverse natural language tasks, from complex reasoning and mathematical problem solving to creative writing and code generation (Chowdhery et al., 2023; Grattafiori et al., 2024; Comanici et al., 2025; Yang et al., 2025). However, deploying large models with billions of parameters requires substantial computational power, which limits their adoption in resource-constrained environments. To this end, knowledge distillation (KD) (Buciluǎ et al., 2006; Hinton et al., 2015; Sanh et al., 2019) has emerged as a promising paradigm to address this challenge by transferring knowledge from large teacher models to smaller, more efficient student models that could preserve most of the teacher capability.

Knowledge distillation methods can be broadly categorized into black-box and white-box approaches. Under black-box KD, the student model learns from training data generated by the teacher model through standard supervised fine-tuning (Kim & Rush, 2016; Taori et al., 2023; Chiang et al., 2023; Xu et al., 2023). Despite its simplicity and broad applicability, black-box KD fails to utilize the rich knowledge embedded in the intermediate representations of the teacher model (Gu et al., 2023; Wen et al., 2023; Ko et al., 2024). Therefore, white-box KD methods aims to further improve student model training by leveraging more information from the teacher, including logit distributions (Sanh et al., 2019; Liang et al., 2023) and internal hidden states (Sun et al., 2019; Jiao et al., 2019;

---

∗ Work done at Amazon Web Services. † Corresponding author: `to58@njit.edu`

Hou et al., 2020; Zuo et al., 2022; Shen et al., 2025). However, applying white-box KD becomes particularly challenging when the teacher and student models use different tokenizers, as token misalignment makes direct knowledge transfer between sequence representations infeasible. To address these challenges, recent work has explored white-box KD for cross-tokenizer scenarios, where not only the model structures but also the vocabularies are different between the teacher and student models (Boizard et al., 2024; Zhang et al., 2025b; Chen et al., 2025; Cui et al., 2025; Minixhofer et al., 2025).

Despite their success, recent work has mainly focused on divergence-based methods that optimize output distribution matching (Xu et al., 2024) without considering scenarios where teacher predictions are incorrect. In this case, the teacher can transfer detrimental bias to student, resulting in poor performance and generalization capacity of the student model (Zhang et al., 2025a). Moreover, *teacher hacking* (Tiapkin et al., 2025) could be another concerning issue, where the student model learns to mimic superficial patterns rather than acquiring meaningful knowledge. These limitations are further compounded by more challenging knowledge distillation settings when teacher and student models are significantly different in the model architectures, e.g. knowledge distillation from a Mixture-of-Experts (MoE) model to a dense model.

To this end, we propose a novel white-box KD framework to address the above challenges. Our approach consists of two stages: pretraining and distillation. In the pretraining stage, we compress teacher hidden states into a low-dimensional space and train the teacher projectors via a self-reconstruction mechanism, which extracts task-relevant information from the teacher model. In the distillation stage, we introduce a novel mechanism called *residual learning*, in which the student learns to predict next tokens using the residual hidden states computed by subtracting projected teacher hidden states from those of student whenever the teacher's predictions are inaccurate. This mechanism encourages the student to learn complementary knowledge that captures what it understands differently from the teacher and avoid replicating teacher errors. To support the cross-tokenizer distillation, we adopt a cross-model attention mechanism, inspired by Zhang et al. (2025b), that automatically establishes the alignments between teacher and student token representations through hidden state similarity. Finally, to better exploit the rich knowledge distributed across experts in a MoE teacher model, we introduce a lightweight scaled dot-production self-attention mechanism that enriches expert outputs before top-$k$ aggregation. These components form a unified framework which is effective for distillation across diverse architectures and tokenization schemes.

We evaluate our framework on standard instruction-following benchmarks following recent works (Gu et al., 2023; Zhang et al., 2025b; Chen et al., 2025) under different distillation settings including cross-tokenizer distillation and distillation from MoE to dense models. Experimental results demonstrate that our method consistently outperforms the existing white-box KD approaches on average Rouge-L scores for different scenarios.

In summary, our contributions are as follows:

- We introduce a two-stage knowledge distillation framework that effectively extracts teacher knowledge for fine-tuning a student model.

- We introduce a residual learning approach that effectively leverages teacher's hidden states and errors as learning signals, enabling student model to generalize better.

- We adopt a cross-model attention mechanism that aligns teacher and student tokens based on their hidden state similarity to handle token mismatches in cross-tokenizer distillation.

- We enhance knowledge extraction for MoE teachers by using a scaled dot-product attention mechanism, fully leveraging all experts' knowledge.

- Experiments show that our method significantly outperforms the existing white-box methods under diverse distillation settings.

## 2 RELATED WORK

**Knowledge Distillation for LLMs**. A straightforward method for knowledge distillation is black-box KD (Kim & Rush, 2016). Let $\mathcal{T}$ and $\mathcal{S}$ represent the teacher and student models, respectively. Black-box KD fine-tunes the student model using the standard causal language modeling objective

on teacher-generated texts:

$$\mathcal{L}_{\text{KD}} = \mathbb{E}_{\mathbf{x} \sim \mathcal{X}, \mathbf{y} \sim \mathbb{P}_{\mathcal{T}}(\cdot|\mathbf{x})} \left[ -\log \mathbb{P}_{\mathcal{S}}(\mathbf{y}|\mathbf{x}) \right], \tag{1}$$

where $\mathbf{y}$ is the output sequence generated by teacher model, $\mathbb{P}_{\mathcal{T}}$ and $\mathbb{P}_{\mathcal{S}}$ are the output probability distributions from the teacher and student, respectively. Although simple, black-box KD has proven to be an effective method in diverse tasks (Wang et al., 2023; Taori et al., 2023; Chiang et al., 2023).

Recent research has shifted to white-box KD to leverage the intermediate information from LLMs beyond their final outputs. Specifically, these methods attempt to align output distributions or maximize the similarity between the hidden states of the teacher and student models (Sanh et al., 2019; Liang et al., 2023; Sun et al., 2019; Zuo et al., 2022; Shen et al., 2025). One common approach is minimizing the divergence between the output probability distributions of the teacher and student:

$$\mathcal{L}_{\text{divergence}} = \mathbb{E}_{(\mathbf{x},\mathbf{y}) \sim \mathcal{X} \times \mathcal{Y}} \left[ D_{\text{KL}} \left( \mathbb{P}_{\mathcal{T}}(\mathbf{y}|\mathbf{x}) \parallel \mathbb{P}_{\mathcal{S}}(\mathbf{y}|\mathbf{x}) \right) \right] \tag{2}$$

where the data point $(\mathbf{x}, \mathbf{y})$ are sampled from the data space $\mathcal{X} \times \mathcal{Y}$ and the closed-form expression of $D_{\text{KL}}$ is determined based on the specific divergence metric adopted. Popular choices for $D_{\text{KL}}$ include Kullback-Leibler (KL) divergence, reverse KL (Gu et al., 2023), Jensen-Shannon divergence (Agarwal et al., 2024), and their skew variants (Lee, 2001; Ko et al., 2024; 2025).

While proven to be effective, divergence-based methods can potentially limit the student's learning capacity. The information from the teacher model is not perfect as it can still make wrong predictions and exhibit certain bias (Zhang et al., 2025a). As a result, forcing the student to mimic the teacher's distributions may propagate such imperfection to the student model and prevent the student from learning beyond the teacher's knowledge boundaries.

**Cross-Tokenizer Knowledge Distillation.** One restriction in white-box KD is that it requires the same vocabulary between the student and the teacher for distribution alignment. Recent work has further explored white-box KD for cross-tokenizer distillation settings, where the student and teacher have different vocabularies. Cross-tokenizer KD setting poses more challenges than conventional same-tokenizer settings, including mismatches in sequence length and output dimensionality. To tackle these challenges, ULD (Boizard et al., 2024) applies simple truncation strategies to align sequence lengths and logit dimensions, enabling direct comparison of output distributions. Based on ULD, MultiLevelOT (Cui et al., 2025) integrates an approximate optimal transport objective to reduce the divergence at both token and sequence levels. Dual-space KD (DSKD) (Zhang et al., 2025b) projects the student's hidden states into the teacher space and vice versa for calculating the divergence between output distributions. It further introduces a cross-model attention mechanism to align tokens across disparate tokenizers using token embeddings similarity. ALM (Minixhofer et al., 2025) further refines token alignment by exhaustively enumerating decoded chunks and identifying equivalent segments between teacher and student outputs.

Despite these advances, existing cross-tokenizer KD methods exhibit several inherent limitations. Truncation-based strategies such as ULD may discard valuable information and suffer from alignment errors. MultiLevelOT improves performance but incurs computational overhead due to optimal transport, limiting its scalability. Projection-based approaches like DSKD rely on KL divergence on the output distributions, which could suffer from the limitations discussed above. Brute-force methods such as ALM achieve precise alignment but scale poorly with large vocabularies or extended sequences. Moreover, none of these state-of-the-art methods address teacher hacking (Tiapkin et al., 2025) or mitigate teacher bias (Zhang et al., 2025a), which could restrict the generalization capacity of the student when the teacher itself is imperfect.

**Distillation from Mixture-of-Experts.** Despite the superior capacity of Mixture-of-Experts (MoE) models on various natural language tasks, few works have specifically explored how to fully distill their inherent power (Salinas et al., 2022; Kim et al., 2025). While Kim et al. (2025) highlighted the importance of incorporating knowledge from all experts during distillation and proposed two mechanisms for expert knowledge extraction, their approaches face practical limitations. The first approach relies on stochastic expert sampling, which requires multiple forward passes to access different expert combinations and increases computational overhead during training. The second approach attempts to adjust router probabilities based on student preferences, but this may result in suboptimal expert selection as it prioritizes the student's current state over the teacher's expertise. These limitations motivate the need for a more efficient and effective method to distill knowledge from MoE teachers while preserving the specialized capabilities of individual experts.

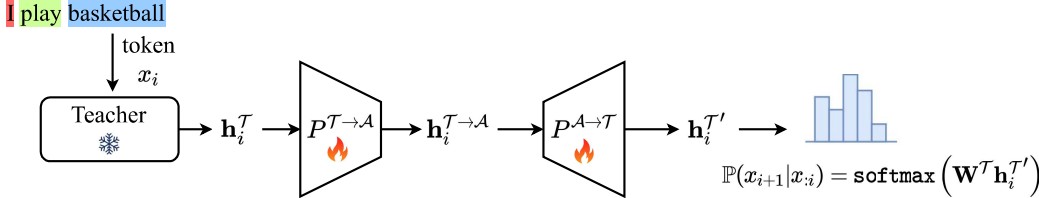

(a) Stage 1: Pretraining teacher projectors. The projectors $P^{\mathcal{T}\to\mathcal{A}}$ and $P^{\mathcal{A}\to\mathcal{T}}$ are learned by optimizing next-token prediction objective using the reconstructed hidden states.

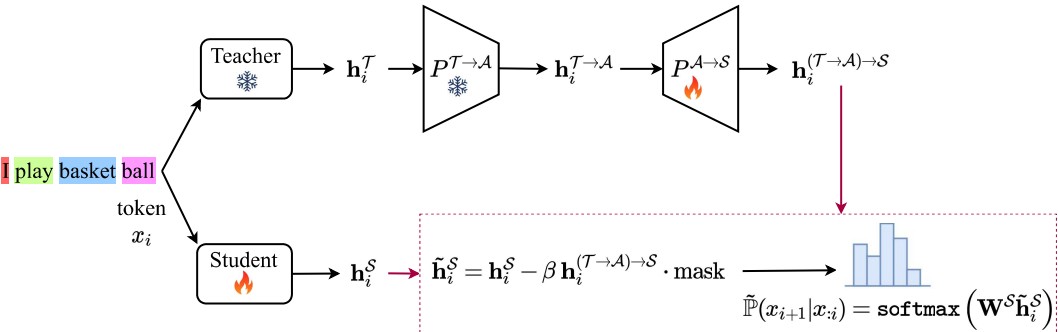

(b) Stage 2: Residual learning objective. We first compute the residual hidden states $\tilde{\mathbf{h}}_i^{\mathcal{S}}$, then use this quantity to obtain the adjusted output probability distribution $\tilde{\mathbb{P}}\left(x_{i+1}|x_{:i}\right)$ for next-token prediction.

Figure 1: Overview of our proposed two-stage framework. Modules marked with fire icons have learnable parameters that are updated during training, while those marked with snow icons have frozen weights.

## 3 METHODOLOGY

In this section, we introduce a two-stage white-box KD framework that is applicable to various KD settings to address the existing limitations. Figure 1 illustrates the overall workflow of our framework. In the first stage, we employ learnable projectors to compress teacher hidden states into a low-dimensional space that is agnostic to any student architecture and the projectors are optimized using a **self-reconstruction** mechanism as shown in Figure 1(a). Section 3.1 describes the details of the first stage. Figure 1(b) illustrates the workflow of the second stage which performs knowledge distillation from the teacher to the model. We introduce a novel mechanism called **residual learning** in Section 3.2 that leverages the residual hidden states calculated from the student hidden states and projected teacher hidden states to effectively guide student learning. Furthermore, we present the **expert knowledge fusion** mechanism in Section 3.3, which utilizes the knowledge from all experts to enhance distillation from MoE teachers. Finally, we introduce a **cross-model attention** mechanism in Section 3.4, aiming at handling sequence mismatch in cross-tokenizer KD settings. The total objective function for distillation is given in Section 3.5.

### 3.1 PRETRAINING PROJECTORS WITH SELF-RECONSTRUCTION

The pretraining stage learns to compress teacher representations while preserving their semantics and task-relevant information. We denote $\mathcal{A}$ as a low-dimensional space whose dimension is $d_{\mathcal{A}}$, and the hidden state dimensions of the student and teacher as $d_{\mathcal{S}}$ and $d_{\mathcal{T}}$, respectively. Let $P^{\mathcal{T}\to\mathcal{A}}$ and $P^{\mathcal{A}\to\mathcal{T}}$ denote the projection functions that map hidden states between $\mathcal{T}$ and $\mathcal{A}$. As shown in Figure 1(a), these projectors form an autoencoder in the self-reconstruction process, where teacher hidden states are compressed into $\mathcal{A}$, and then reconstructed. The reconstructed hidden states $\mathbf{h}_i^{\mathcal{T}'}$ are subsequently used for next-token prediction, and the projectors are optimized via the cross-entropy objective:

$$\mathcal{L}_{\text{CE}} = -\sum_i \log \mathbb{P}(x_{i+1}|x_{:i}), \quad \text{with} \quad \mathbb{P}\left(x_{i+1}|x_{:i}\right) = \texttt{softmax}\left(\mathbf{W}^{\mathcal{T}}\mathbf{h}_i^{\mathcal{T}'}\right) \tag{3}$$

where $\mathbf{W}^{\mathcal{T}}$ is the teacher's prediction head. We denote $\mathbf{h}_i^{\mathcal{T} \to \mathcal{A}}$ as the compressed teacher hidden states, *i.e.*, $\mathbf{h}_i^{\mathcal{T} \to \mathcal{A}} = P^{\mathcal{T} \to \mathcal{A}} \mathbf{h}_i^{\mathcal{T}}$. In the subsequent stage, $P^{\mathcal{T} \to \mathcal{A}}$ remains frozen to stabilize teacher information during distillation.

## 3.2 RESIDUAL LEARNING

Recent work highlights two key challenges in distillation: *teacher hacking*, where the student exploits superficial patterns in the teacher's outputs rather than learning meaningful knowledge (Tiapkin et al., 2025), and *teacher bias*, where prediction errors are directly transferred to the student. To address these challenges, we propose *residual learning* for LLM distillation, which leverages not only the projected hidden states from the teacher but also the teacher errors as informative signals to guide student learning.

Let $\mathbf{h}_i^{(\mathcal{T} \to \mathcal{A}) \to \mathcal{S}}$ denote the projected teacher hidden states of token $i$ that are mapped to the student space $\mathcal{S}$, *i.e.*, $\mathbf{h}_i^{(\mathcal{T} \to \mathcal{A}) \to \mathcal{S}} = P^{\mathcal{A} \to \mathcal{S}} \mathbf{h}_i^{\mathcal{T} \to \mathcal{A}}$. We compute the **residual hidden states** by taking the difference between the student hidden states and the projected teacher representations in the student space at the positions where the teacher makes incorrect predictions:

$$\tilde{\mathbf{h}}_i^{\mathcal{S}} = \mathbf{h}_i^{\mathcal{S}} - \beta \mathbf{h}_i^{(\mathcal{T} \to \mathcal{A}) \to \mathcal{S}} \cdot \mathbb{1} \left[ \arg\max \mathbb{P}_{\mathcal{T}}(x_i | x_{:i-1}) \neq x_i \right], \tag{4}$$

where $\tilde{\mathbf{h}}_i^{\mathcal{S}}$ represents the residual hidden states, $\beta$ is a scaling factor, and the indicator function equals 1 only when the teacher's top-1 prediction is different from the ground-truth token. The scaling factor $\beta$ is crucial for balancing the contribution of teacher and student in the residual term. If $\beta$ is too large, student representations are overwhelmed by those of teachers, making the residuals ineffective. In contrast, if $\beta$ is too small, the residual terms approximate the student hidden states, and the benefits of residual learning would diminish. Therefore, we adaptively compute proper $\beta$ values according to the dimensions and magnitudes of the hidden states as below:

$$\beta = \underbrace{\sqrt{\frac{d_{\mathcal{S}}}{d_{\mathcal{A}}}}}_{\text{rescale dimension}} \times \underbrace{\frac{1}{n} \sum_{i=1}^{n_{\mathcal{S}}} \frac{\left\| \mathbf{h}_i^{\mathcal{S}} \right\|}{\left\| \mathbf{h}_i^{(\mathcal{T} \to \mathcal{A}) \to \mathcal{S}} \right\|}}_{\text{rescale magnitude}} \tag{5}$$

where $n_{\mathcal{S}}$ stands for the student's sequence length. The first component in $\beta$ normalizes for dimensional differences between spaces, while the second term aligns the magnitudes of teacher and student representations to ensure balanced residual computation. These ratios prevent either the student or teacher representations from dominating the residual calculation. We empirically observe that using $\beta$ at sequence level (average of token-wise norm ratios) can stabilize training and yield better results.

Then, we train the student model to make accurate predictions based on the residual hidden states using the student's LM head $\mathbf{W}^{\mathcal{S}}$ with the cross-entropy loss:

$$\mathcal{L}_{\text{res}} = -\sum_{i=1}^{n_{\mathcal{S}}} \log \tilde{\mathbb{P}} \left( x_{i+1} | x_{:i} \right), \quad \text{where} \quad \tilde{\mathbb{P}} \left( x_{i+1} | x_{:i} \right) = \texttt{softmax} \left( \mathbf{W}^{\mathcal{S}} \tilde{\mathbf{h}}_i^{\mathcal{S}} \right) \tag{6}$$

This objective fundamentally changes the learning dynamics: instead of passively replicating the teacher's distributions, the student learns to actively identify the differences between its current understanding and the teacher's knowledge. This prevents the superficial pattern matching that characterizes teacher imperfection, ultimately leading to better generalization of the student model.

## 3.3 EXPERT KNOWLEDGE FUSION

For teacher models with MoE structures, we propose an efficient expert knowledge fusion mechanism to effectively utilize the knowledge of all experts for better knowledge distillation. Given $E$ experts in the teacher model, we denote $\mathbf{h}_i^{(m)}$ as the outputs of the $m$-th expert at the token position $i$ for the last MoE layer. We enrich the expert outputs through scaled dot-product attention:

$$\tilde{\mathbf{h}}_i^{(m)} = \sum_{j=1}^{E} \alpha_{mj} \mathbf{h}_i^{(j)}, \quad \text{where} \quad \alpha_{mj} = \texttt{softmax} \left( \frac{\mathbf{h}_i^{(m)} (\mathbf{h}_i^{(j)})^{\top}}{\sqrt{d_{\mathcal{T}}}} \right) \tag{7}$$

where $\alpha_{mj}$ captures the relevance between the outputs of the $m$-th and $j$-th experts. This self-attention mechanism allows each expert to incorporate complementary information from similar experts, resulting in more informative representations. Finally, the enriched expert outputs are aggregated to form the final states following the original MoE workflow:

$$\mathbf{h}_i^{\mathcal{T}} = \sum_{j \in \text{top-}k} g_j \tilde{\mathbf{h}}_i^{(j)} \tag{8}$$

where $g_j$ denotes the router probability assigned to the $j$-th expert, and top-$k$ denotes the $k$ experts selected by the router. Unlike previous methods that rely on stochastic sampling or router probability adjustment (Kim et al., 2025), our attention-based fusion can directly leverage the complementary knowledge of all experts in a single forward pass.

### 3.4 CROSS-MODEL ATTENTION MECHANISM

In Section 3.2, we introduce our residual learning design in the same-tokenizer setting where the input sequences for the teacher and student are exactly the same. In the cross-tokenizer setting, since teacher and student have different vocabularies, their tokenizers will output sequences of different lengths for the same text input. This hinders the application of residual learning.

To address this challenge, we propose a cross-model attention mechanism to align the compressed teacher hidden states $\mathbf{h}_i^{\mathcal{T} \rightarrow \mathcal{A}}$ for the input tokens of the student. We first map the student hidden states $\mathbf{h}_i^{\mathcal{S}}$ to the same low-dimensional space $\mathcal{A}$ (denoted as $\mathbf{h}_i^{\mathcal{S} \rightarrow \mathcal{A}}$) via a trainable projector. Then, for each pair of student token $i$ and teacher token $j$, we compute their semantic similarity via dot-products using the normalized hidden states, followed by row-wise `softmax` normalization:

$$\mathbf{A}_{ij} = \left[ \frac{\mathbf{h}_i^{\mathcal{S} \rightarrow \mathcal{A}}}{\text{std}\left(\mathbf{h}_i^{\mathcal{S} \rightarrow \mathcal{A}}\right)} \right]^{\top} \left[ \frac{\mathbf{h}_j^{\mathcal{T} \rightarrow \mathcal{A}}}{\text{std}\left(\mathbf{h}_j^{\mathcal{T} \rightarrow \mathcal{A}}\right)} \right], \quad \text{then} \quad \mathbf{A}[i,:] = \text{softmax}\left( \frac{\mathbf{A}[i,:]}{\sqrt{d_{\mathcal{A}}}} \right) \tag{9}$$

By computing similarity scores between all token pairs, we construct a cross-model attention matrix $\mathbf{A} \in \mathbb{R}^{n_{\mathcal{S}} \times n_{\mathcal{T}}}$, where $n_{\mathcal{S}}$ and $n_{\mathcal{T}}$ are the sequence lengths for student and teacher respectively. Each element $\mathbf{A}_{ij}$ represents the semantic similarity between student token $i$ and teacher token $j$. Using the cross-model attention matrix $\mathbf{A}$, we calculate the aligned teacher hidden states $\hat{\mathbf{h}}_i^{\mathcal{T} \rightarrow \mathcal{A}}$ for each student token by taking a weighted sum of the compressed teacher hidden states:

$$\hat{\mathbf{h}}_i^{\mathcal{T} \rightarrow \mathcal{A}} = \sum_{j=1}^{n_{\mathcal{T}}} \mathbf{A}_{ij} \mathbf{h}_j^{\mathcal{T} \rightarrow \mathcal{A}}, \tag{10}$$

This mechanism allows each student token to attend to the most semantically relevant teacher tokens, effectively bridging the tokenization gap and preserving semantic information. Then, we can use $\hat{\mathbf{h}}_i^{\mathcal{T} \rightarrow \mathcal{A}}$ instead to perform residual learning under the cross-tokenizer settings.

### 3.5 TRAINING OBJECTIVE

We incorporate the residual learning objective from Eqn. 6 with a standard supervised fine-tuning (SFT) objective where the student model learns to predict the next token from the ground-truth data. The final loss function is:

$$\mathcal{L} = \lambda \mathcal{L}_{\text{res}} + (1 - \lambda) \mathcal{L}_{\text{SFT}}, \tag{11}$$

where $\lambda$ is a hyperparameter that balances the contribution of each loss component.

## 4 EXPERIMENTS

### 4.1 EXPERIMENTAL SETUP

**Models.** We evaluate our framework under both the same-tokenizer and cross-tokenizer settings. For the same-tokenizer setting, we conduct experiments on knowledge distillation from Mixtral-8×7B-Instruct (Jiang et al., 2024) to Mistral-7B and distillation from LLaMA2-7B (Touvron et al., 2023) to TinyLLaMA-1.1B (Zhang et al., 2024). To optimize LLaMA2-7B's performance as a teacher model,

we further fine-tune it on the training dataset before knowledge distillation. For the cross-tokenizer setting, we use Mixtral-8×7B-Instruct as the teacher model and use TinyLLaMA-1.1B and GPT2-120M (Radford et al., 2019) as the student models. We perform full fine-tuning on GPT2-120M and LoRA fine-tuning on TinyLLaMA-1.1B and Mistral-7B. In this way, our evaluation covers diverse knowledge distillation settings in practice.

**Datasets.** For training, we use the Dolly dataset (Ouyang et al., 2022), processed by Gu et al. (2023), which comprises approximately 11,000 training samples and 1,000 validation samples. We evaluate our method on five standard instruction-following benchmarks: Dolly (500 test samples), Self-Instruction (Wang et al., 2023), Vicuna-Eval (Chiang et al., 2023), Super-Natural Instructions (Wang et al., 2022), and Unnatural Instruction (Honovich et al., 2023). We adopt Rouge-L scores (%) as the primary metric across all datasets.

**Baselines.** We select the following approaches as baselines: supervised fine-tuning (SFT), Universal Logits Distillation (ULD, Boizard et al. (2024)), Multi-Level Optimal Transport (MultiLevelOT, Cui et al. (2025)), Dual-space KD (DSKD, (Zhang et al., 2025b)), and Approximate Likelihood Matching (ALM, (Minixhofer et al., 2025)). Although these baselines were specifically designed for cross-tokenizer KD, they are also capable of achieving state-of-the-art performance in the same-tokenizer setting, except for ALM which can only be applied for cross-tokenizer KD. We additionally include ABKD (Wang et al., 2025), which leverages $\alpha$-$\beta$-divergence to improve distillation, as a same-tokenizer baseline.

**Implementation Details.** We conduct experiments using PyTorch Distributed on NVIDIA A100 GPUs (40GB). We use `bfloat16` precision and gradient accumulations to manage memory constraints. The dimension of the space $\mathcal{A}$ is set as $d_{\mathcal{A}} = 64$ across all experiments, and all projector modules are linear weights without bias. Teacher projectors are pretrained for 10 epochs as described in Section 3.1 and we use the checkpoint at the final step. For final evaluation, we select the checkpoint with the best validation Rouge-L score during distillation. The detailed hyperparameters can be found in Appendix A.

## 4.2 RESULTS

**Same-tokenizer KD results.** Table 1 reports the Rouge-L scores for same-tokenizer KD across multiple benchmarks. For *Mixtral-8×7B-Instruct → Mistral-7B*, our method achieves the highest average score (30.68), which matches the teacher zero-shot performance (30.67) and significantly outperforms the best baseline MultiLevelOT (29.16). For *LLaMA2-7B → TinyLLaMA-1.1B*, our method surpasses all baselines, reaching an average score of 25.17, which is $+0.62$ above the best baseline (DSKD). These results demonstrate the effectiveness of our proposed framework in improving student's generalization, potentially matching its teacher's performance.

**Cross-tokenizer KD results.** Table 2 presents the results for cross-tokenizer KD settings. For *Mixtral-8×7B-Instruct → TinyLLaMA-1.1B*, our method again achieves the best average performance (25.09), outperforming the best baseline DSKD (23.89) by $+1.20$. For *Mixtral-8×7B-Instruct → GPT2-120M*, our approach reaches 20.01 average Rouge-L, significantly outperforming all baselines. We also note that the performance gap between the teacher and the fine-tuned student is larger in the cross-tokenizer settings, which is due to the the compounded difficulty of cross-tokenizer alignment and smaller student model size.

In summary, for both same- and cross-tokenizer KD, our method outperforms the existing baselines. The improvements are robust across datasets and teacher-student pairs, highlighting the effectiveness of our approach in preserving and transferring knowledge.

## 4.3 ABLATION STUDY

To understand the effectiveness of each design choice in our framework, we conduct comprehensive ablation studies under the cross-tokenizer setting where we use Mixtral-8×7B-Instruct as the teacher and GPT2-120M as the student.

**Effectiveness of each technique.** We consider the following variants of our method: 1) "w/o $\beta$", in which we remove $\beta$ for computing the residual hidden states; 2) "w/o accuracy mask", in which we remove the indicator function for computing the residual hidden states; 3) "w/o pretrained $P^{\mathcal{T} \rightarrow \mathcal{A}}$",

Table 1: Performance comparison using Rouge-L scores (%) for same-tokenizer KD. We report the average scores over 3 random seeds. "Avg." is the average score across five instruction-following benchmarks.

| Methods | Dolly | SelfInst | VicunaEval | S-NI | UnNI | Avg. |
|---|---|---|---|---|---|---|
| Mixtral-8×7B-Instruct → Mistral-7B | | | | | | |
| Teacher zero-shot | 25.55 | 24.12 | 26.62 | 40.84 | 36.21 | 30.67 |
| Student SFT | 25.85 | 22.17 | 17.18 | 33.41 | 30.26 | 25.77 |
| ULD | 28.64 | 21.99 | 21.31 | 35.71 | 34.42 | 28.41 |
| MultiLevelOT | 30.55 | 23.07 | 20.62 | 34.79 | 36.78 | 29.16 |
| DSKD | 25.35 | 20.43 | 18.27 | 35.39 | 31.45 | 26.18 |
| ABKD ($\alpha = \beta = 0.5$) | 29.53 | 24.11 | **21.80** | 37.31 | 36.56 | 29.86 |
| Ours | **31.39** | **24.27** | 20.48 | **38.82** | **38.42** | **30.68** |
| LLaMA2-7B → TinyLLaMA-1.1B | | | | | | |
| Teacher SFT | 29.82 | 20.65 | 19.18 | 30.66 | 33.11 | 26.68 |
| Student SFT | 24.26 | 16.41 | 16.13 | 25.70 | 27.41 | 21.98 |
| ULD | 26.13 | 17.52 | 16.82 | 28.56 | 29.34 | 23.67 |
| MultiLevelOT | 24.51 | 15.75 | 16.25 | 25.72 | 26.00 | 21.65 |
| DSKD | **26.66** | 18.05 | 17.61 | 30.11 | 30.30 | 24.55 |
| ABKD ($\alpha = \beta = 0.5$) | 26.57 | 17.18 | 16.90 | 30.25 | 30.93 | 24.37 |
| Ours | 26.10 | **18.11** | **17.97** | **31.13** | **32.54** | **25.17** |

Table 2: Performance comparison using Rouge-L scores (%) for cross-tokenizer KD. We report the average scores over 3 random seeds.

| Methods | Dolly | SelfInst | VicunaEval | S-NI | UnNI | Avg. |
|---|---|---|---|---|---|---|
| Teacher zero-shot | 25.55 | 24.12 | 26.62 | 40.84 | 36.21 | 30.67 |
| Mixtral-8×7B-Instruct → TinyLLaMA-1.1B | | | | | | |
| Student SFT | 24.26 | 16.41 | 16.13 | 25.70 | 27.41 | 21.98 |
| ULD | 23.61 | 15.45 | 17.06 | 29.01 | 28.45 | 22.71 |
| MultiLevelOT | 22.52 | 14.48 | 14.97 | 27.06 | 25.77 | 20.96 |
| ALM | 21.93 | 14.11 | 15.37 | 25.75 | 25.49 | 20.53 |
| DSKD | **26.16** | 18.02 | **18.15** | 27.06 | 30.08 | 23.89 |
| Ours | 25.84 | **19.38** | 17.35 | **30.86** | **32.04** | **25.09** |
| Mixtral-8×7B-Instruct → GPT2-120M | | | | | | |
| Student SFT | 21.45 | 9.45 | 14.89 | 16.86 | 19.14 | 16.36 |
| ULD | 22.12 | 10.54 | 15.03 | 18.16 | 20.12 | 17.19 |
| MultiLevelOT | 21.40 | 9.57 | 15.56 | 15.88 | 18.05 | 16.09 |
| ALM | 21.16 | 9.77 | 15.46 | 16.09 | 18.26 | 16.15 |
| DSKD | **23.11** | 11.20 | **15.81** | 20.33 | 21.56 | 18.40 |
| Ours | 22.62 | **11.66** | 15.12 | **24.51** | **26.15** | **20.01** |

in which we remove the pretraining stage and train $P^{\mathcal{T} \rightarrow \mathcal{A}}$ directly during distillation; 4) "w/o MoE expert fusion", in which we remove our expert knowledge fusion design. We further consider another variant to leverage the knowledge of all experts which simply activates and aggregates all the experts, i.e. using top-$k$ as top-8 for Mixtral-8×7B-Instruct. Table 3 compares the results of our method with these variants. We can find that all these variants perform significantly worse than the original method, which demonstrates the effectiveness of each design. Among them, removing the coefficient $\beta$ leads to the most significant performance drop, highlighting its role in balancing the teacher–student contributions within residual learning.

Table 3: Ablation study results for different variants under the knowledge distillation from Mixtral-8×7B-Instruct (teacher) to GPT2-120M (student).

| Description | Dolly | SelfInst | VicunaEval | S-NI | UnNI | Avg. |
|---|---|---|---|---|---|---|
| Mixtral-8×7B-Instruct → GPT2-120M | | | | | | |
| Ours | 22.62 | 11.66 | 15.12 | 24.51 | 26.15 | 20.01 |
| w/o $\beta$ | 22.31 | 9.59 | 14.43 | 16.01 | 18.93 | 16.25 |
| w/o accuracy mask | 22.63 | 11.04 | 14.79 | 24.02 | 25.45 | 19.59 |
| w/o pretraining $P^{\mathcal{T} \rightarrow \mathcal{A}}$ | 21.52 | 10.07 | 15.24 | 18.81 | 21.31 | 17.39 |
| w/o freezing pretrained $P^{\mathcal{T} \rightarrow \mathcal{A}}$ | 22.33 | 10.46 | 15.39 | 23.21 | 23.98 | 19.07 |
| w/o MoE expert fusion | | | | | | |
|    – sparse activation (2/8 experts) | 22.68 | 11.84 | 14.82 | 23.36 | 25.61 | 19.66 |
|    – dense activation (8/8 experts) | 23.09 | 10.52 | 16.10 | 23.88 | 24.82 | 19.68 |
|    – average pooling (8/8 experts) | 22.90 | 11.17 | 13.52 | 21.11 | 23.43 | 18.43 |

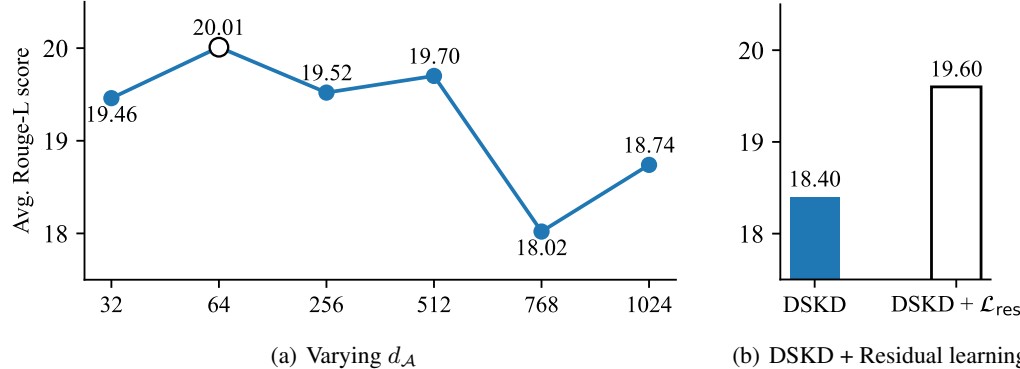

(a) Varying $d_{\mathcal{A}}$

(b) DSKD + Residual learning

Figure 2: Cross-tokenizer distillation results on (a) varying the dimensionality of the space $\mathcal{A}$ (b) incorporating residual learning into DSKD.

**Impact of the low-dimensional space.** Under the cross-tokenizer settings, the low-dimensional space $\mathcal{A}$ is used to compress teacher hidden states and perform cross-model attention for alignment. Here, we further study the impact of the dimension $d_{\mathcal{A}}$ on the distillation performance. Figure 2(a) shows the distillation performance when varying $d_{\mathcal{A}}$. We find that increasing $d_{\mathcal{A}}$ does not provide additional benefit and a very large dimension (e.g. $d_{\mathcal{A}} = 768$ or $1024$) can even significantly harm the performance. It proves that compressing teacher hidden states into a low dimension is not only beneficial but essential for effective knowledge transfer. In our case, $d_{\mathcal{A}} = 64$ yields the best result (20.01), indicating a balance between compression and expressivity.

**Generality of residual learning.** To demonstrate the effectiveness of residual learning, we further apply our design into other white-box KD methods. Figure 2(b) compares the performance of incorporating the residual loss $\mathcal{L}_{\text{res}}$ with the baseline DSKD. We see an improvement of $+1.20$ on average after incorporating residual learning. It demonstrates the generality of our residual learning mechanism, which has the potential to improve the performance of other white-box KD methods.

## 5 CONCLUSION

In this work, we introduce a novel two-stage KD framework that addresses key challenges arising from the imperfect teacher knowledge in white-box KD settings. We compress the teacher hidden states into a low-dimension space prior to distillation and guide student training through residual learning, which ensures robust knowledge transfer and effectively mitigates overfitting to the teacher's imperfection. Our framework further provides targeted designs for diverse KD settings in-

cluding cross-tokenizer distillation and distillation from an MoE model. Extensive experiments on multiple instruction-following benchmarks demonstrate that our method consistently outperforms the existing baselines and significantly improves the capacity of the student model under different settings. In the future, we will further investigate if our method can generalize to other tasks such as reasoning-intensive tasks and code generation.

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

## A    CONFIGURATIONS AND HYPERPARAMETERS

In Table 4, we provide details of the configurations and hyperparameters for training and evaluation.

Table 4: Training configuration and hyperparameters

|  | Parameter | Value |
|---|---|---|
| **Hardware & Training** | GPU | $8\times$ NVIDIA A100 (40 GB) |
|  | Global batch size | 128 |
|  | Max sequence length | 512 |
|  | Precision | `bfloat16` |
| **Optimization** | Optimizer | AdamW |
|  | Learning rate (student) | $10^{-3}$ |
|  | Learning rate (projectors) | $10^{-3}$ |
|  | Learning rate scheduler | Cosine decay |
|  | Weight decay | $10^{-4}$ |
|  | Epochs | 10 |
|  | Loss coefficient ($\lambda$) | 0.5 |
| **Architecture** | Alignment space dim ($d_{\mathcal{A}}$) | 64 |
|  | Projector layers | Linear (without bias) |
|  | Teacher projector pretraining | 10 epochs |
| **LoRA Parameters** | Modules | $Q, O,$ Gate/Up/Down projections |
|  | Rank | 256 |
|  | $\alpha$ | 8 |
|  | Dropout rate | 0.1 |
| **Sampling & Generation** | Temperature | 1.0 |
|  | Top P | 1.0 |

## B    THEORETICAL FRAMEWORK FOR RESIDUAL LEARNING

In this section, we provide theoretical results explaining why the proposed *residual learning* mechanism, which subtracts a projected teacher state during training, is effective. Theorems 1 formalizes the key principle: the residual term induces an implicit regularizer that pushes the student away from teacher directions on teacher-error tokens. We then analyze the convergence of residual learning by gradient descent in Theorem 2.

For readability, we denote $\mathbf{u}_i = \mathbf{h}_i^{(\mathcal{T}\to\mathcal{A})\to\mathcal{S}}$ as the mapped teacher hidden states. The residual hidden states defined in Eq. 4 can be rewritten as follows:

$$\tilde{\mathbf{h}}_i^{\mathcal{S}} = \mathbf{h}_i^{\mathcal{S}} - \beta\mathbf{u}_i \cdot \mathbb{1}\left[\arg\max \mathbb{P}_{\mathcal{T}}(x_i|x_{:i-1}) \neq x_i\right] \tag{12}$$

We also denote the student's model parameters as $\boldsymbol{\theta}$ and its LM head as $\mathbf{W}$. Hence, the student logits and the residual logits can be defined as $\mathbf{z}_i := \mathbf{W}\mathbf{h}_i^{\mathcal{S}}$ and $\tilde{\mathbf{z}}_i = \mathbf{W}\tilde{\mathbf{h}}_i^{\mathcal{S}}$, respectively. Finally, the cross-entropy loss function is represented by $\ell(\cdot)$.

**Theorem 1** (Implicit Regularization Against Teacher-Error Directions)**.** *For any token $i$ on which the teacher prediction is incorrect, the gradient of $\mathcal{L}_{\mathrm{res}}$ with respect to $\boldsymbol{\theta}$ satisfies*

$$\nabla_{\boldsymbol{\theta}}\mathcal{L}_{\mathrm{res}} = \nabla_{\boldsymbol{\theta}}\mathcal{L}_{\mathrm{SFT}} - \beta(\mathbf{J}_i^{\mathcal{S}})^{\top}\bar{\mathbf{H}}_i\mathbf{u}_i, \tag{13}$$

*where:*

- $\mathcal{L}_{\mathrm{SFT}}$ *is the standard cross-entropy loss using logits* $\mathbf{z}_i = \mathbf{W}\mathbf{h}_i^{\mathcal{S}}$,

- $\mathbf{J}_i^{\mathcal{S}} = \dfrac{\partial \mathbf{h}_i^{\mathcal{S}}}{\partial \boldsymbol{\theta}}$ *is the student Jacobian,*

- $\mathbf{H}_i = \dfrac{\partial^2 \ell(\mathbf{z}_i)}{\partial \mathbf{z}_i \partial \mathbf{z}_i^{\top}}$ *is the Hessian matrix of the cross-entropy loss, and* $\bar{\mathbf{H}}_i := \mathbf{W}^{\top}\mathbf{H}_i\mathbf{W}$.

*Thus residual learning adds an extra term that regularizes the gradient updates of student model parameters, pushing the gradients against the teacher's hidden states direction $\mathbf{u}_i$ on tokens where the teacher predicts incorrectly.*

*Proof.* For any position $i$-th where teacher exhibits error and the corresponding ground truth token $x_i$, the residual loss can be expressed as:

$$\mathcal{L}_{\text{res}} = \mathbb{E}_i \left[\ell(\mathbf{W}\tilde{\mathbf{h}}_i^{\mathcal{S}}, x_i)\right] = \mathbb{E}_i \left[\ell(\tilde{\mathbf{z}}_i, x_i)\right]$$

By chain rule:

$$\nabla_{\mathbf{h}_i^{\mathcal{S}}} \mathcal{L}_{\text{res}} = \underbrace{\frac{\partial \mathbf{z}_i}{\partial \mathbf{h}_i^{\mathcal{S}}}}_{=\mathbf{W}^\top} \cdot \underbrace{\frac{\partial \tilde{\mathbf{z}}_i}{\partial \mathbf{z}_i}}_{=I} \cdot \frac{\partial \ell(\tilde{\mathbf{z}}_i, x_i)}{\partial \tilde{\mathbf{z}}_i}$$

$$= \mathbf{W}^\top \nabla_{\mathbf{z}_i} \ell(\tilde{\mathbf{z}}_i, x_i)$$

Perform a first-order Taylor expansion of $\nabla_{\mathbf{z}_i} \ell(\tilde{\mathbf{z}}_i)$ around $\mathbf{W}\mathbf{h}_i^{\mathcal{S}}$:

$$\nabla_{\mathbf{z}_i} \ell(\mathbf{W}\mathbf{h}_i^{\mathcal{S}} - \beta \mathbf{W}\mathbf{u}_i) = \nabla_{\mathbf{z}_i} \ell(\mathbf{W}\mathbf{h}_i^{\mathcal{S}}) - \beta \mathbf{H}_i \mathbf{W}\mathbf{u}_i + o(\beta)$$

Multiplying both sides by $\mathbf{W}^\top$, we have:

$$\nabla_{\mathbf{h}_i^{\mathcal{S}}} \mathcal{L}_{\text{res}} = \nabla_{\mathbf{h}_i^{\mathcal{S}}} \mathcal{L}_{\text{SFT}} - \beta \mathbf{W}^\top \mathbf{H}_i \mathbf{W}\mathbf{u}_i$$

$$= \nabla_{\mathbf{h}_i^{\mathcal{S}}} \mathcal{L}_{\text{SFT}} - \beta \bar{\mathbf{H}}_i \mathbf{u}_i, \quad \text{where } \bar{\mathbf{H}}_i = \mathbf{W}^\top \mathbf{H}_i \mathbf{W}$$

Finally, chaining derivatives through parameters $\boldsymbol{\theta}_{\mathcal{S}}$ yields Eq. 13. The equation shows that the residual term induces regularization $\left(-\beta (\mathbf{J}_i^{\mathcal{S}})^\top \bar{\mathbf{H}}_i \mathbf{u}_i\right)$ during gradient update, pushing the gradients against the teacher's hidden states direction on teacher-error tokens. $\square$

**Convergence Analysis.** We now show that the residual objective used in our method guarantees convergence under mild smoothness assumptions. To simplify notations, we remove the subscript $i$ in previous proofs, denote $f_{\boldsymbol{\theta}}(x) \in \mathbb{R}^{d_{\mathcal{S}}}$ as the student representation for input token $x$, $\mathbf{h}_{\mathcal{T}} \in \mathbb{R}^{d_{\mathcal{A}}}$ as the compressed teacher hidden states, and $g_{\boldsymbol{\phi}} : \mathbb{R}^{d_{\mathcal{A}}} \mapsto \mathbb{R}^{d_{\mathcal{S}}}$ as the projector that maps compressed teacher states to student space parameterized by $\boldsymbol{\phi}$. The logits and losses are:

$$\mathbf{z} = \mathbf{W}f_{\boldsymbol{\theta}}(x), \quad \tilde{\mathbf{z}} = \mathbf{W}\left[f_{\boldsymbol{\theta}}(x) - \beta g_{\boldsymbol{\phi}}(\mathbf{h}_{\mathcal{T}})\right], \quad \mathcal{L}_{\text{SFT}} = \mathbb{E}[\ell(\mathbf{z}, x)], \quad \mathcal{L}_{\text{res}} = \mathbb{E}[\ell(\tilde{\mathbf{z}}, x)]$$

and the combined distillation objective is:

$$\mathcal{L}(\boldsymbol{\theta}, \boldsymbol{\phi}) = \mathcal{L}_{\text{SFT}}(\boldsymbol{\theta}) + \lambda \mathcal{L}_{\text{res}}(\boldsymbol{\theta}, \boldsymbol{\phi})$$

Note that the combined objective defined here is different from Eq. 11, w.l.o.g. We consider the update of both the student parameters $\boldsymbol{\theta}$ and the projector parameters $\boldsymbol{\phi}$ by gradient descent:

$$\boldsymbol{\theta}_{t+1} = \boldsymbol{\theta}_t - \eta_{\boldsymbol{\theta}} \nabla_{\boldsymbol{\theta}} \mathcal{L}(\boldsymbol{\theta}_t, \boldsymbol{\phi}_t), \quad \boldsymbol{\phi}_{t+1} = \boldsymbol{\phi}_t - \eta_{\boldsymbol{\phi}} \nabla_{\boldsymbol{\phi}} \mathcal{L}(\boldsymbol{\theta}_t, \boldsymbol{\phi}_t)$$

We make the following standard assumptions that control the step of gradient descent:

(A1) The cross-entropy $\ell(\mathbf{z}, x)$ is $L_{\ell}$-Lipschitz and $L_z$-smooth in the logits $\mathbf{z}$.

(A2) The gradient of $\ell$ w.r.t. logits is bounded: $\|\nabla_{\mathbf{z}} \ell(\mathbf{z}, x)\| \leq B_g$.

(A3) For each fixed $\boldsymbol{\phi}$, $\mathcal{L}(\boldsymbol{\theta}, \boldsymbol{\phi})$ is $L_{\boldsymbol{\theta}}$-smooth in $\boldsymbol{\theta}$; for each fixed $\boldsymbol{\theta}$, $\mathcal{L}(\boldsymbol{\theta}, \boldsymbol{\phi})$ is $L_{\boldsymbol{\phi}}$-smooth in $\boldsymbol{\phi}$.

(A4) For each $\boldsymbol{\phi}$ in the region of interest, $\mathcal{L}(\boldsymbol{\theta}, \boldsymbol{\phi})$ is $\mu_{\boldsymbol{\theta}}$-strongly convex in $\boldsymbol{\theta}$.

(A5) The mapping $g_{\boldsymbol{\phi}}$ is $L_g$-Lipschitz in $\boldsymbol{\phi}$: $\|g_{\boldsymbol{\phi}_{t+1}}(\mathbf{h}_{\mathcal{T}}) - g_{\boldsymbol{\phi}_t}(\mathbf{h}_{\mathcal{T}})\| \leq L_g \|\boldsymbol{\phi}_{t+1} - \boldsymbol{\phi}_t\|$.

(A6) The student LM head has bounded norm: $\|\mathbf{W}\| \leq \sigma$, or $\|\mathbf{W}\| = \sigma$ for simplicity.

**Lemma 1** (Bounded Drift From Projector Update.). *We define $\Delta_t := \mathcal{L}(\boldsymbol{\theta}_t, \boldsymbol{\phi}_{t+1}) - \mathcal{L}(\boldsymbol{\theta}_t, \boldsymbol{\phi}_t)$ as the "drift" in total loss function caused by updating $\boldsymbol{\phi}$ at iteration t. If $\mathcal{L}(\boldsymbol{\theta}, \boldsymbol{\phi})$ is $L_{\boldsymbol{\phi}}$-smooth in $\boldsymbol{\phi}$, the drift is bounded as follows:*

$$|\Delta_t| \leq \eta_{\boldsymbol{\phi}} \lambda \beta \sigma B_g L_g \|\nabla_{\boldsymbol{\phi}} \mathcal{L}(\boldsymbol{\theta}_t, \boldsymbol{\phi}_t)\|$$

*Proof.* Let $d_t := \phi_{t+1} - \phi_t$ and $\phi(s) = \phi_t + sd_t$ for $s \in [0, 1]$. Then:

$$
\begin{aligned}
|\Delta_t| &= \int_0^1 \langle \nabla_\phi \mathcal{L}(\boldsymbol{\theta}_t, \phi(s)), d_t \rangle ds \\
&\leq \int_0^1 \|d_t\| \cdot \|\nabla_\phi \mathcal{L}(\boldsymbol{\theta}_t, \phi(s))\| \, ds \quad \text{(by Cauchy-Schwarz)} \\
&= \|d_t\| \int_0^1 \|\nabla_\phi \mathcal{L}(\boldsymbol{\theta}_t, \phi(s))\| \, ds \\
&\leq \|d_t\| \sup_{s \in [0,1]} \|\nabla_\phi \mathcal{L}(\boldsymbol{\theta}_t, \phi(s))\| \\
&= \|\phi_{t+1} - \phi_t\| \sup_{\tilde{\phi}} \left\|\nabla_\phi \mathcal{L}(\boldsymbol{\theta}_t, \tilde{\phi})\right\| \\
&= \eta_\phi \|\nabla_\phi \mathcal{L}(\boldsymbol{\theta}_t, \phi_t)\| \sup_{\tilde{\phi}} \left\|\nabla_\phi \mathcal{L}(\boldsymbol{\theta}_t, \tilde{\phi})\right\|
\end{aligned}
$$

From the residual logits $\tilde{\mathbf{z}} = \mathbf{W} f_{\boldsymbol{\theta}_t}(x) - \lambda \beta \mathbf{W} g_\phi(\mathbf{h}_\mathcal{T})$, the chain rule gives:

$$
\nabla_\phi \mathcal{L}(\boldsymbol{\theta}_t, \phi) = \mathbb{E}\left[ (\mathbf{J}_\phi g_\phi(\mathbf{h}_\mathcal{T}))^\top (-\lambda \beta \mathbf{W})^\top \nabla_{\mathbf{z}} \ell(\tilde{\mathbf{z}}, x) \right]
$$

where $\mathbf{J}_\phi g_\phi(\mathbf{h}_\mathcal{T})$ is the Jacobian of $g_\phi$ at $\mathbf{h}_\mathcal{T}$. Taking norms and applying Jensen's inequality, the Lipschitz property of $g_\phi$, the bound $\|\mathbf{W}\| = \sigma$, and Assumption A2,

$$
\begin{aligned}
\|\nabla_\phi \mathcal{L}(\boldsymbol{\theta}_t, \phi)\| &\leq \lambda \beta \sigma \, \mathbb{E}\left[\|\nabla_{\mathbf{z}} \ell(\tilde{\mathbf{z}}, x)\| \|\mathbf{J}_\phi g_\phi(\mathbf{h}_\mathcal{T})\|\right] \\
&\leq \lambda \beta \sigma B_g L_g
\end{aligned}
$$

Therefore, $\sup_{\tilde{\phi}} \left\|\nabla_\phi \mathcal{L}(\boldsymbol{\theta}_t, \tilde{\phi})\right\| = \lambda \beta \sigma B_g L_g$. Substituting this supremum term to bound $|\Delta_t|$ will complete the proof. $\qquad \square$

**Theorem 2** (Convergence of Residual Learning). *Suppose Assumptions A1–A6 hold. If $\eta_{\boldsymbol{\theta}} \leq 1/L_{\boldsymbol{\theta}}$ and the projector learning rate $\eta_\phi$ is sufficiently small, then the gradient descent update of the total loss function $\mathcal{L}(\boldsymbol{\theta}, \phi)$ satisfies:*

$$
\mathcal{L}(\boldsymbol{\theta}_{t+1}, \phi_{t+1}) - \mathcal{L}^* \leq (1 - \eta_{\boldsymbol{\theta}} \mu_{\boldsymbol{\theta}}) [\mathcal{L}(\boldsymbol{\theta}_t, \phi_t) - \mathcal{L}^*] + |\Delta_t| \tag{14}
$$

*where $\mathcal{L}^* = \inf_{\boldsymbol{\theta}, \phi} \mathcal{L}(\boldsymbol{\theta}, \phi)$ is the optimal loss value and $|\Delta_t|$ is bounded as in Lemma 1. Specifically, for moderate $\lambda, \beta$ and sufficiently small $\eta_\phi$, the drift term is small and the total loss $\mathcal{L}(\boldsymbol{\theta}, \phi)$ decreases and converges to a neighborhood of $\mathcal{L}^*$.*

*Proof.* Fix $\phi_t$ and consider the update in $\boldsymbol{\theta}$. By $L_{\boldsymbol{\theta}}$-smoothness:

$$
\begin{aligned}
\mathcal{L}(\boldsymbol{\theta}_{t+1}, \phi_t) &= \mathcal{L}(\boldsymbol{\theta}_t - \eta_{\boldsymbol{\theta}} \nabla_{\boldsymbol{\theta}} \mathcal{L}(\boldsymbol{\theta}_t, \phi_t), \phi_t) \\
&\leq \mathcal{L}(\boldsymbol{\theta}_t, \phi_t) - \eta_{\boldsymbol{\theta}} \|\nabla_{\boldsymbol{\theta}} \mathcal{L}(\boldsymbol{\theta}_t, \phi_t)\|^2 + \frac{L_{\boldsymbol{\theta}} \eta_{\boldsymbol{\theta}}^2}{2} \|\nabla_{\boldsymbol{\theta}} \mathcal{L}(\boldsymbol{\theta}_t, \phi_t)\|^2
\end{aligned}
$$

Using $\eta_{\boldsymbol{\theta}} \leq 1/L_{\boldsymbol{\theta}}$ and $\mu_{\boldsymbol{\theta}}$-strong convexity in $\boldsymbol{\theta}$, the Polyak–Łojasiewicz inequality gives

$$
\|\nabla_{\boldsymbol{\theta}} \mathcal{L}(\boldsymbol{\theta}_t, \phi_t)\|^2 \geq 2\mu_{\boldsymbol{\theta}} [\mathcal{L}(\boldsymbol{\theta}_t, \phi_t) - \mathcal{L}^*]
$$

Therefore,

$$
\mathcal{L}(\boldsymbol{\theta}_{t+1}, \phi_t) - \mathcal{L}^* \leq (1 - \eta_{\boldsymbol{\theta}} \mu_{\boldsymbol{\theta}}) [\mathcal{L}(\boldsymbol{\theta}_t, \phi_t) - \mathcal{L}^*]
$$

Now we consider the update of $\phi$, which induces a "drift" term $\Delta_t$ in total loss:

$$
\mathcal{L}(\boldsymbol{\theta}_{t+1}, \phi_{t+1}) - \mathcal{L}^* \leq (1 - \eta_{\boldsymbol{\theta}} \mu_{\boldsymbol{\theta}}) [\mathcal{L}(\boldsymbol{\theta}_t, \phi_t) - \mathcal{L}^*] + |\Delta_t|
$$

where $|\Delta_t|$ is bounded as in Lemma 1. With moderate $\lambda$ and $\beta$, it requires that the projector learning rate $\eta_\phi$ remains slow to maintain a reasonably low magnitude of $|\Delta_t|$, which is critical to guarantee convergence toward the neighborhood of the optimal value $\mathcal{L}^*$. $\qquad \square$

## C    ANALYSIS: RESIDUAL LEARNING VS. DUAL SPACE KD UNDER DIVERGENCE VARIANTS

To further evaluate the robustness of our framework, we compare residual learning with the strong Dual-Space KD (DSKD) baseline under different divergence loss functions. In particular, we replace the KL divergence in DSKD with alternatives including reverse KL, Jensen–Shannon (JS) divergence, adaptive KL, and skewed KL variants, and measure the student performance in each setting. All experiments are conducted with Mixtral-8×7B-Instruct as the teacher and GPT2-120M as the student. The results in Table 5 show that residual learning consistently outperforms DSKD across all divergence objectives, underscoring its generalization and effectiveness. By leveraging the teacher's errors as learning signals, residual learning prevents the student from imitating the teacher's output distributions in a rigid manner, resulting in stronger generalization.

Table 5: Comparing residual learning vs. DSKD under divergence variants.

| Description | Dolly | SelfInst | VicunaEval | S-NI | UnNI | Avg. |
|---|---|---|---|---|---|---|
| Mixtral-8×7B-Instruct → GPT2-120M | | | | | | |
| DSKD + forward KL | 23.11 | 11.20 | 15.81 | 20.33 | 21.56 | 18.40 |
| DSKD + reverse KL | 22.03 | 9.44 | 12.95 | 17.08 | 18.41 | 15.98 |
| DSKD + adaptive KL | 23.41 | 10.65 | 14.76 | 18.88 | 20.46 | 17.63 |
| DSKD + JS divergence | 22.60 | 10.59 | **16.21** | 18.44 | 20.37 | 17.64 |
| DSKD + skew forward KL | **23.63** | 10.89 | 15.57 | 19.42 | 21.21 | 18.14 |
| DSKD + skew reverse KL | 23.15 | 11.10 | 15.64 | 20.47 | 21.67 | 18.40 |
| **Ours** | 22.62 | **11.66** | 15.12 | **24.51** | **26.15** | **20.01** |

## D    SENSITIVITY ANALYSIS: LOSS COEFFICIENT $\lambda$

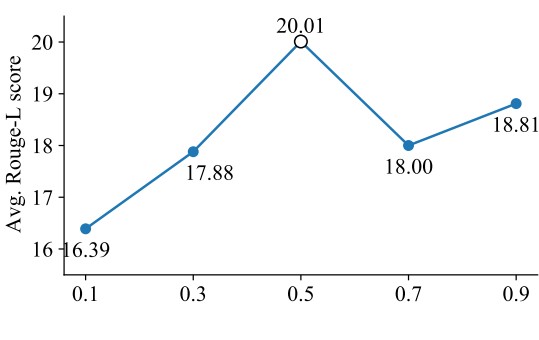

Figure 3: Varying $\lambda$

Figure 3 illustrates the sensitivity of our method w.r.t. the loss coefficient $\lambda$ (from Eq. 11). We evaluate the performance using Mixtral-8×7B-Instruct as the teacher and GPT2-120M as the student. We vary the value of $\lambda \in \{0.1, 0.3, 0.5, 0.7, 0.9\}$) and report the average Rouge-L scores over 3 random seeds. The highest score of $20.01$ is achieved at $\lambda = 0.5$, indicating that this value provides the most effective weighting between the SFT and residual objectives. Extremely small or large values of $\lambda$ degrade performance, suggesting that both components of the loss are essential.

## E    SAMPLE PROMPTS AND ANSWERS

We investigate the outputs of distillation from LLaMA2-7B → TinyLLaMA-1.1B and show several prompts and outputs where the teacher model and baseline methods predict incorrectly at some token position, and residual learning can learn to avoid replicating the same mistakes. These examples demonstrate that residual learning has the capability to correct teacher's mistakes.

---
**Example 1 – Token-level correction**

**Prompt:** Below is an instruction that describes a task, paired with an input that provides further context. Write a response that appropriately completes the request.
Instruction: You need to output the text of the passage after capitalizing all proper nouns.

---

Input: Passage: Rami went to new york city on business. While he was there, he met his good friend ali, who showed him around town.
**Ground truth:** Rami went to New York City on business. While he was there, he met his good friend Ali, who showed him around town.
**Teacher output:** Rami went to New York City on business. While he was there, he met his Good Friend Ali, who showed him around **Town**.
**DSKD output:** Passage: Rami wentto new york city onbusiness. While he was there , he met his good friend, ali, who showed him around **Town**. Passage: Rams, ali.
**Ours:** Rami went to New York city on business while he was there he met his good friend Ali who showed him around **town**.

---

**Example 2 – Sequence-level correction**

**Prompt:** Below is an instruction that describes a task, paired with an input that provides further context. Write a response that appropriately completes the request.
Instruction: For this task, you will be given a question and two pieces of text. Your job is to determine which piece of text answers the question. The answer should be one or more phrases in the paragraph, and it should be unambiguous.
Input: Question: What phrase best completes the sentence "He was pleased with himself for ____"?
Text 1: He had defeated his opponent handily in the arm-wrestling match.
Text 2: He had finally completed his ten thousandth pushup.
**Ground truth:** He was pleased with himself for defeating his opponent handily in the arm-wrestling match.
**Teacher output:** The correct answer is "he finally completed his ten thousand pushup". This is because when someone says he is pleased with himself it usually refers to a task or accomplishment that he is proud of. In this case, "he" in the question could refer to either "He had defeated his opponent" or "He had finally completed his ten thousand pushup" and we want to identify the accomplishment that he is actually pleased with. Since the question is worded as "what phrase completes the sentence", the entire phrase needs to fit in the sentence, therefore the answer is "he had finally completed his ten thousands pushup" (hallucination, wrong choice.)
**DSKD output:** In both texts, He was pleased with himself for getting handily defeated in the arm-wristling match. He had completed his ten thousandth pushing up. (hallucination, wrong choice.)
**Ours:** He was pleased with himself for winning the arm-wrestler match. (correct answer).
*In this example, we can observe that the teacher and DSKD tends to hallucinate while residual learning can avoid making the same mistakes.*

