# OpenReview forum: "Knowledge Distillation for Large Language Models through Residual Learning"
_ICLR.cc/2026/Conference — ICLR 2026 Poster_

### Official Review · Reviewer_SYtA · 2025-10-29

**Soundness:** 2
**Presentation:** 2
**Contribution:** 2
**Rating:** 2
**Confidence:** 4

**Summary:**

The paper presents a two-stage white-box knowledge distillation framework. In the first stage, the teacher’s hidden states are compressed into a low-dimensional latent space via self-reconstruction projectors. In the second stage, the student is trained through a residual learning mechanism, where it refines its representations by subtracting the teacher’s hidden states from its own. The paper claims that this design encourages the student to learn complementary knowledge and mitigate the propagation of teacher errors. Experimental results on instruction-following benchmarks demonstrate consistent performance gain.

**Strengths:**

- The introduction of a feature distillation method where the student learns from the difference between its and the teacher’s hidden states.

- The paper is well written and structured.

- KD for LLM is a very important topic.

**Weaknesses:**

- Projecting the teacher’s hidden states into the student space is an established paradigm widely used in feature-based distillation to handle mismatched hidden dimensions [1,2,3].

- The evaluation is restricted to instruction-following tasks and a limited set of model architectures. Experiments on more diverse tasks such as reasoning, summarization, etc are important.

- The paper does not include comparisons with several recent and relevant knowledge distillation methods (e.g., DistillM, DistillM2, ABKD, CKA-based distillation), limiting the strength of its empirical claims.

- The paper motivates residual learning as a means mitigate teacher errors, yet the link between this motivation and the proposed residual mechanism is not clear.



[1] Jiao X, Yin Y, Shang L, Jiang X, Chen X, Li L, Wang F, Liu Q. Tinybert: Distilling bert for natural language understanding. arXiv preprint arXiv:1909.10351. 2019 Sep 23.

[2] Miles, Roy, and Krystian Mikolajczyk. "Understanding the role of the projector in knowledge distillation." Proceedings of the AAAI Conference on Artificial Intelligence. Vol. 38. No. 5. 2024.

[3] Chen, Yudong, et al. "Improved feature distillation via projector ensemble." Advances in Neural Information Processing Systems 35 (2022): 12084-12095.

**Questions:**

See weaknesses section

---

> ### Author Response · Authors · 2025-12-03
>
> We thank the reviewer for providing useful feedback. We would like to present our answers to your questions as below.
>
> > Projecting the teacher’s hidden states into the student space is an established paradigm widely used in feature-based distillation to handle mismatched hidden dimensions [1, 2, 3].
>
> We acknowledge that these references made significant contributions to distillation, but their applicability in LLM distillation---especially for generative models such as decoder-only transformers---remains unclear. For instance, [1] focuses on aligning hidden states for encoder-only model (BERT), [2] and [3] attempts to align feature maps in vision applications.
>
> Furthermore, our contribution first maps the teacher hidden states to a low dimensional space, agnostic of any student models. Although this mapping might have been used by previous works, we successfully apply it into decoder-only LLM distillation, making our novel contribution.
>
> > Experiments on more diverse tasks such as reasoning, summarization, etc are important.
>
> We acknowledge that code and math reasoning tasks are important to assess the effectiveness of the methods. However, the instruction following tasks we included has been known to contain a large part of problems that require a certain level of reasoning, as used in previous works (Dual-space KD, MiniLLM, DistiLLM, DistiLLM-2, etc.). In fact, many previous works only used these 5 benchmarks to report their final distillation results. For math and code tasks, they require specialized teacher and student models that have been pretrained on these specific tasks to compare the methods reliably. We will consider these extra tasks in our future work using a different set of teacher and student models.
>
> > comparisons with several recent and relevant knowledge distillation methods (e.g., DistillM, DistillM2, ABKD, CKA-based distillation)
>
> We identify ABKD (alpha-beta-divergence) [4] as a relevant approach for comparing with our method (same-tokenizer only). To the best of our knowledge, the other suggested baselines are not very relevant to our work. For instance, the CKA-based methods are applied for vision model distillation, while DistiLLM-2 are not based on a supervised dataset. We add the new baseline ABKD to Table 1 and show that our method still outperforms ABKD by +0.8 in average score as shown in Table 1.
>
> > the link between this motivation and the proposed residual mechanism is not clear.
>
> To clarify the reviewer’s question, we would like to summarize the motivation and proposed framework as follows.
>
> Our motivation for introducing residual learning stems from a fundamental limitation of existing white-box distillation methods: when the teacher is imperfect, directly aligning the student to its hidden states or output distributions can propagate the teacher’s errors and biases. This is especially problematic because these errors tend to be systematic, and a student trained to imitate them will inherit the same failure modes rather than surpass the teacher.
>
> Residual learning was designed precisely to address this challenge. Instead of treating all teacher signals as equally reliable, our mechanism explicitly distinguishes between trustworthy and untrustworthy teacher predictions. The key idea is that, on tokens where the teacher is wrong, the teacher’s representation should not serve as a target to imitate, but as a direction to move away from. This is where our residual term comes into play.
>
> In other words, the residual mechanism is not merely a modification of the hidden states—it is a direct instantiation of our core motivation: to leverage teacher knowledge selectively, and to actively counteract teacher errors rather than absorb them. This behavioral distinction is further supported by our theoretical analysis (Appendix C), where we show that the residual term induces an implicit regularizer that biases the student away from aligning with teacher-error directions.
>
>
> \textbf{References}
>
> [1] Jiao X, Yin Y, Shang L, Jiang X, Chen X, Li L, Wang F, Liu Q. Tinybert: Distilling bert for natural language understanding. arXiv preprint arXiv:1909.10351. 2019 Sep 23.
>
> [2] Miles, Roy, and Krystian Mikolajczyk. "Understanding the role of the projector in knowledge distillation." Proceedings of the AAAI Conference on Artificial Intelligence. Vol. 38. No. 5. 2024.
>
> [3] Chen, Yudong, et al. "Improved feature distillation via projector ensemble." Advances in Neural Information Processing Systems 35 (2022): 12084-12095.
>
> [4] Wang, Guanghui, et al. “ABKD: Pursuing a proper allocation of the probability mass in knowledge distillation via divergence.” In Proceedings of the 42nd International Conference on Machine Learning, 2025

---

### Official Review · Reviewer_4ahb · 2025-10-31

**Soundness:** 3
**Presentation:** 3
**Contribution:** 3
**Rating:** 6
**Confidence:** 4

**Summary:**

Traditional knowledge distillation may forces the student model to inherit the errors of an imperfect teacher. This paper proposes a novel framework called "Residual Learning" to address the core issue. The key idea is to guide the student to learn only the "difference"(residual) between its own understanding and its teacher, but only when the teacher makes a mistake.Thereby avoiding blind imitation and holding the potential to surpass the teacher.
Combined with specialized modules designed for Mixture-of-Experts (MoE) models and cross-tokenizer scenarios, experiments show that this framework significantly outperforms existing methods across various distillation tasks and effectively enhances the generalization ability of the student model.

**Strengths:**

● Clear Problem Definition and Motivation:This paper identifies a issue in white-box knowledge distillation: the teacher model is not perfect. Forcing the student model to align with the output distribution of an imperfect teacher essentially sets an upper bound on its capability and may propagate biases. The proposed idea of "critical learning instead of blind imitation" is highly persuasive and innovative.
● Novel, Intuitive, and Easy-to-Understand Core Method:The method uses an indicator function 1[teacher is wrong] to determine when to activate "residual learning", shifting the focus from imitation to error correction. The idea is both intuitive and easy to understand.
● Comprehensive Experiments and Strong Results:
● Broad Range of Scenarios Covered:The experiments span various realistic and challenging distillation scenarios, including same/cross-tokenizer settings and distilling from Mixture-of-Experts (MoE) models to dense models.
● Solid Ablation Studies:The ablation study is well-designed and thoroughly evaluates the importance of each component in residual learning: the accuracy mask, scaling factor β, pre-trained projector, and MoE expert fusion. This significantly strengthens the credibility of the proposed method.

**Weaknesses:**

1. In scenarios where there are no definitive correct answers, such as text generation or chain-of-thought (CoT) reasoning, the indicator function for determining whether the teacher is wrong may be limited in its applicability.
2. It would greatly enhance the persuasiveness of the paper if there are concrete examples that demonstrate the core claim of this paper — for instance, showing the process where "the teacher makes a mistake → the baseline student model follows the error → the proposed student model successfully corrects it"

**Questions:**

● Regarding the dependence on ground-truth labels:Do you think it is possible to extend the idea of "residual learning" to scenarios where no ground-truth is available? For example, could one use a discriminator model or leverage the consistency among multiple teacher models as a proxy signal for whether the teacher made a mistake?
● Regarding the breadth of evaluation:Have you considered evaluating your method on tasks that emphasize logic rather than text matching, such as reasoning, math problems, or code generation? We are particularly interested in whether residual learning over token-level prediction errors can effectively transfer and enhance complex reasoning capabilities.
● Regarding hyperparameter sensitivity:The final loss is a weighted combination of lambda​. In the experiments, both set to 0.5. How sensitive is the model performance to the ratio between these two coefficients? Have you conducted any sensitivity analysis regarding this?
● Regarding the MoE fusion mechanism:In the MoE knowledge fusion, you employ self-attention to aggregate outputs from all experts. Compared to simpler alternatives like averaging all expert outputs (average pooling) or activating all experts (dense activation), how much improvement does the self-attention mechanism actually bring? This comparison does not seem to be explicitly included in the ablation study.

---

> ### Author Response · Authors · 2025-12-03
>
> We appreciate the reviewer for providing constructive feedback, which is very useful for understanding and further clarifying the mechanism and effectiveness of our framework. Please see our answers to your questions/comments as below.
>
> > In scenarios where there are no definitive correct answers, such as text generation or chain-of-thought (CoT) reasoning, the indicator function for determining whether the teacher is wrong may be limited in its applicability.
>
> The benchmark we’re experimenting with is based on instruction-following, which doesn’t have definitive correct answers. For scenarios where multiple reasoning paths can lead to the same correct answers, we will experiment with more tasks in our future work.
>
> > concrete examples that demonstrate the core claim of this paper — for instance, showing the process where "the teacher makes a mistake → the baseline student model follows the error → the proposed student model successfully corrects it"
>
> We include several samples to show that residual learning can avoid making the same mistakes as the teacher model and other baselines. We choose DSKD, the best baseline, to show case the comparisons in Appendix E.
>
> > Do you think it is possible to extend the idea of "residual learning" to scenarios where no ground-truth is available?
>
> As long as a token predicted by the teacher is identified as wrong, we believe the effectiveness of residual learning still holds regardless of the existence of ground truth labels. We will consider the evaluation of this scenario in future work.
>
> >  Have you considered evaluating your method on tasks that emphasize logic rather than text matching, such as reasoning, math problems, or code generation?
>
> We will consider these tasks using more specialized models in our future work.
>
> > In the experiments, both were set to 0.5. How sensitive is the model performance to the ratio between these two coefficients?
>
> We add appendix D in the revised version, where we vary $\lambda \in \{0.1, 0.3, 0.5, 0.7, 0.9\}$ and found that it is important to balance the effect of both loss components, with the optimal performance achieved at $\lambda = 0.5$. These results show that some tuning will be required to balance the loss components.
>
> > Compared to simpler alternatives like averaging all expert outputs (average pooling) or activating all experts (dense activation), how much improvement does the self-attention mechanism actually bring?
>
> We already compare MoE expert fusion with other simple baselines such as (i) dense activation and (ii) sparse activation using the original router probability. In the revised version, we also include average pooling of all experts (uniform router probability) and show that our expert fusion mechanism still yields superior performance. As shown in Table 3, MoE expert fusion yields 20.01 average Rouge-L score, outperforming sparse activation (19.66), dense activation (19.68), and average pooling of all experts (18.43).

---

### Official Review · Reviewer_hWgu · 2025-10-31

**Soundness:** 3
**Presentation:** 3
**Contribution:** 4
**Rating:** 8
**Confidence:** 4

**Summary:**

This paper presents a method for knowledge distillation including several novel innovations, notably "residual learning", a method to guide the student in order to avoid replicating mistakes from the teacher, among other innovations around effective combination of learning signal from MoE teachers, and handling teacher-student combinations with different tokenization methods.

Overall the paper was really interesting, the goals lofty, and the approach very engaging. My one gripe is I'm not sure I understood why residual learning works, and how the overall training objective can benefit positively from the student as it appears only to include negative supervision when the teacher makes mistakes. Hopefully the authors can clarify in the discussion phase.

**Strengths:**

This paper goes beyond the norm in presenting more algorithmic innovations than most. The residual part is the core contribution -- and ablations showed this to work very effectively -- but the MoE and cross-tokenizer methods were also good additions, slightly simpler in formulation, but very sensible and shown to have utility.

The experimental evaluation is solid, showing the utility of the method and its component parts on standard evals with a range of strong models (Mistral, LLama etc) at reasonable sizes 7B, and against competitive baselines. In many cases the student outperforms the teacher, despite using many fewer parameters, which is a very impressive result. I would expect these results to be interesting to the wider community.

**Weaknesses:**

The presentation is a bit murky at times. I was left puzzled about why the method works, and what the core motivation is around the various steps in the residual learning method. Let me elaborate on my understanding. Please correct my misconceptions in your rebuttal.

Stage 1: pretraining projectors, this appears to be done to as a form of regularization to smooth out meaningless variation in the teacher hidden states. It also allows for situations where the hidden dimension is different in the student, or the representations are the same size and have similar information but indices are permuted (e.g., identifiability). I'm not sure why it's trained with a softmax/categorical distribution to predict the gold next token - wouldn't matching teacher predictions, e.g., using KL, be more suitable?

Stage 2: during KD the model learns to project the smoothed teacher state to match the student state, and uses this only when there's a teacher mistake, and it fails to correctly predict the gold next token. In these settings, the student's hidden state is offset to subtract the effect of the teacher. Can this be understood as correcting a likely mistake in the student? And given this 'correction', how does learning via backpropagation of loss from the prediction of the gold next token affect the parameters? I see W^S may be better estimated, but I wonder about h^S_i and h^{(T -> A) -> S}_i and how they will be updated. Given this offsetting of hidden states won't be applied during inference, it's hard to see how this change in KD training will make an impact on the final model.

**Questions:**

Presentation clarification questions, see Weaknesses.

Why does unfreezing teacher projectors has such a negative impact on accuracy?

Why is there no KD loss using the teacher in a positive manner, e.g., KL against teacher outputs, or alignment to the teacher hidden states when the teacher is correct?

In 3.3, eq 8 how is the h_i^T used? Is this a drop in replacement for a dense h_i^T in section 3.2 eqs 4-5?

---

> ### Author Response · Authors · 2025-12-03
>
> We appreciate the feedback the reviewer shared. We would like to address the questions and concerns you have raised.
>
> > why it's trained with a softmax/categorical distribution to predict the gold next token - wouldn't matching teacher predictions, e.g., using KL, be more suitable?
>
> Training with a softmax/categorical distribution to predict gold next token should be more beneficial than matching with teacher's original outputs through KL, since the teacher's outputs can be imperfect, undermining student performance.
>
> > how does learning via backpropagation of loss from the prediction of the gold next token affect the parameters? I see $W^S$ may be better estimated, but I wonder about $h^S_i$ and $h^{(T -> A) -> S}_i$ and how they will be updated. Given this offsetting of hidden states won't be applied during inference, it's hard to see how this change in KD training will make an impact on the final model.
>
> We add Appendix C, where we theoretically prove that the subtraction term in calculating residual hidden states induces an implicit regularizer during training, pushing student model’s gradients away from teacher’s representation on teacher-error tokens. Although the hidden-state offset does not happen during inference, the regularization induced by residual learning in the training phase is reflected in student model parameter updates. Since residual loss is uncommon, we also provide formal analysis to show that the total loss function can converge with gradient descent with convexity assumptions.
>
> > Why does unfreezing teacher projectors have such a negative impact on accuracy?
>
> In the proposed setting (main experiments), we pretrain $P^{T → A}$ in stage 1 and freeze its weights in stage 2. To verify the necessity of pretraining teacher projectors in stage 1 and freezing its weights in stage 2, we present two different ablation studies in Table 3.
>
> (i)  in the experiment “w/o pretraining $P^{T → A}$”, we skip stage 1 and train the projector from scratch during stage 2.
>
> (ii) in the experiment “w/o freezing pretrained $P^{T → A}$”, we pretrain the projector in stage 1, but do not freeze its weights in stage 2. This is the newly added experiment to verify the effect of freezing projector weights.
>
> In both cases (i) and (ii), we observe lower performance compared with our proposed setting. These empirical results demonstrate that our pretraining and freezing strategy can help improve the distillation process: stage 1 improves the projectors, while the weight freezing in stage 2 provides stabilized signals for the student model to learn from.
>
> > Why is there no KD loss using the teacher in a positive manner, e.g., KL against teacher outputs, or alignment to the teacher hidden states when the teacher is correct?
>
> We conduct experiments where KL divergence loss is added where the teacher is correct. However, the empirical results do not support incorporating this KL divergence loss component. Please see the following table for two teacher–student pairs.
>
> | **Loss function** | Dolly | SelfInst | VicunaEval | S-NI | UnNI | Avg. |
> |-------------------|-------|----------|-------------|------|-------|------|
> | **LLaMA2-7B → TinyLLaMA-1.1B (same tokenizer)**|||||||
> | SFT + Residual            | 26.10 | 18.11 | 17.97 | 31.13 | 32.54 | 25.17 |
> | SFT + Residual + KL Div   | 25.59 | 18.28 | 17.23 | 30.02 | 32.36 | 24.70 |
> |**Mixtral-8x7B-Instruct → GPT2-120M (cross tokenizer)**|||||||
> | SFT + Residual            | 22.62 | 11.66 | 15.12 | 24.51 | 26.15 | 20.01 |
> | SFT + Residual + KL Div   | 22.33 | 10.24 | 14.95 | 21.68 | 24.24 | 18.69 |
>
> We suspect adding a new loss component will alter the training dynamic, hence requires careful tuning (e.g., loss coefficient). Nevertheless, KL Divergence on teacher-correct token is promising and we will consider this in our future work.
>
>
>
> > In 3.3, eq 8 how is the $h_i^T$ used? Is this a drop in replacement for a dense $h_i^T$ in section 3.2 eqs 4-5?
>
> Yes, that is how we used it. We will provide clearer description of this quantity in the revised version.

---

### Comment · Area_Chair_A9Hx · 2025-12-01

Dear Authors,

I hope you are doing well. I am writing as the newly assigned area chair for your submission.

This is just a gentle reminder that the author response period remains open until **December 3, 2025, 09:00 PM UTC**. If you wish to provide a rebuttal or additional clarifications addressing the reviewers’ comments, you may still submit them before the deadline.

**Your responses will be carefully considered when preparing the meta-review**, especially given the unusual circumstances of this year’s review process.

Please feel free to reach out if any issues arise.
Thank you very much for your understanding and cooperation.

Warm regards,

AC

---

> ### Author Response · Authors · 2025-12-03
>
> Dear Area Chair,
>
> We have uploaded the revised version of the paper, along with a response addressing the reviewers’ feedback and questions. Thank you very much for overseeing the review of our submission.
>
> Best regards,
>
> The Authors

---

### Meta-Review · Area_Chair_A9Hx · 2025-12-05

**Summary:**

This submission proposes a residual-learning–based LLM distillation framework that selectively corrects teacher errors, supported by an additional projector pretraining stage and extensions for MoE teachers and cross-tokenizer settings. The method is clearly motivated and demonstrates strong empirical gains across multiple realistic distillation scenarios.

However, the concerns raised by Reviewer SYtA, while somewhat high-level and not always tightly grounded in technical specifics, do point to genuine limitations: the paper’s empirical scope remains narrow (instruction following only), comparisons with some recent baselines are incomplete, and the connection between the proposed residual mechanism and the stated motivation could be clearer. The authors’ rebuttal addresses these issues, but mostly at a surface level without fully resolving the conceptual or empirical gaps.

**Reviewer Concerns:**

### **Concerns that were addressed**

* **Clarification of residual mechanism:**
  Authors expanded the explanation, added theoretical analysis, and described how residual offsets induce regularization during training.
* **MoE fusion & cross-tokenizer details:**
  Additional ablations (e.g., average pooling baseline) were included.
* **Projector pretraining / freezing:**
  Ablations clarify why freezing improves stability.
* **Examples of error correction:**
  Added illustrative samples in Appendix E.

These additions improve clarity and help readers understand the methodological intuition.

### **Concerns that remain partially unresolved**

* **Limited evaluation scope:**
  Only instruction-following tasks are used. No reasoning, math, coding, or broader generative settings.
  Reviewers agree this limits generality.
* **Baselines not fully representative:**
  Some relevant KD approaches (e.g., CKA-based, DistillM families) remain untested; authors provide explanations but do not fully eliminate the concern.
* **Residual learning motivation vs. implementation:**
  Although clarified, the conceptual link between teacher error detection and hidden-state subtraction is still not fully convincing to all reviewers.
* **Reviewer SYtA’s critique:**
  While their concerns are somewhat general and lack depth, the authors’ replies also remain mostly qualitative and do not fully settle the underlying questions.

**Reviewer Scores:**

### **Reviewer hWgu (current: 8 → likely remains 8)**

This reviewer is strongly positive about the paper’s contributions, highlighting novelty, empirical strength, and practical impact. Their only major concern was conceptual clarity regarding why residual learning works. The authors provided theoretical explanations (Appendix C), clarified the mechanism, and improved the intuition.
**Given the reviewer’s already high score, the rating would likely remain at 8.**

---

### **Reviewer 4ahb (current: 6 → likely remains 6, with slight positive shift possible)**

This reviewer is largely supportive, finding the idea intuitive, experiments solid, and ablations convincing. Their questions concern broader applicability, concrete examples, and sensitivity analysis. The rebuttal responds to all points and adds missing ablations/samples.

While the reviewer expresses clear interest in the method, the remaining concerns (generalization to reasoning tasks, reliance on ground truth) are structural and cannot be fully resolved within rebuttal constraints.
**Thus, the reviewer would likely maintain a 6.**

---

### **Reviewer SYtA (current: 2 → likely upward shift to 4)**

This reviewer’s concerns centered on novelty, limited empirical scope, and missing baselines. Their review is comparatively high-level and somewhat generic, and although the authors addressed these points (adding ABKD, clarifying projector novelty, and explaining the limitations around reasoning tasks), the responses do not fundamentally change the reviewer’s stance on the paper’s scope.

That said, several of the reviewer’s criticisms were softened by the rebuttal, and some misunderstandings were resolved (e.g., applicability of prior projector-based methods).

---

### Decision · Program_Chairs · 2026-01-26

Accept (Poster)